



# Determination of the atmospheric lifetime and global warming potential of sulphur hexafluoride using a three-dimensional model

Tamás Kovács[1], Wuhu Feng[1,2], Anna Totterdill[1], John M.C. Plane[1], Sandip Dhomse[2],

Juan Carlos Gómez-Martín[1], Gabriele P. Stiller[3], Florian J. Haenel[3], Christopher Smith[4],

Piers M. Forster[2], Rolando R. García[5], Daniel R. Marsh[5] and Martyn P. Chipperfield[2]*

[1]School of Chemistry, University of Leeds, Leeds LS2 9JT, UK.

[2]NCAS, School of Earth and Environment, University of Leeds, Leeds LS2 9JT, UK.

[3]Karlsruhe Institute of Technology, IMK-ASF, PO BOX 3640, 76021 Karlsruhe, Germany.

[4]Energy Research Institute, School of Chemical and Process Engineering, University of Leeds,

Leeds LS2 9JT, UK.

[5]National Center for Atmospheric Research (NCAR), Boulder, Colorado, USA.

*Correspondence to*: Martyn Chipperfield (M.Chipperfield@leeds.ac.uk)

**Abstract.** We have used the Whole Atmosphere Community Climate Model (WACCM), with an updated treatment of loss processes, to determine the atmospheric lifetime of $SF_6$. The model includes the following $SF_6$ removal processes: photolysis, electron attachment and reaction with mesospheric metal atoms. The Sodankylä Ion Chemistry (SIC) model is incorporated into the standard version of WACCM to produce a new version with a detailed $D$ region ion chemistry with cluster ions and negative ions. This is used to determine a latitude- and altitude-dependent scaling factor for the electron density in the standard WACCM in order to carry out multi-year $SF_6$ simulations. The model gives a mean $SF_6$ lifetime over a 11-year solar cycle ($\tau$) of 1278 years (with a range from 1120 to 1475 years), which is much shorter than the currently widely used value of 3200 years, due to the larger contribution (97.4%) of the modelled electron density to the total atmospheric loss. The loss of $SF_6$ by reaction with mesospheric metal atoms (Na and K) is far too slow to affect the lifetime. We investigate how this shorter atmospheric lifetime impacts the use of $SF_6$ to derive stratospheric age-of-air. The age-of-air derived from this shorter lifetime $SF_6$ tracer is longer by 9% in polar latitudes at 20 km compared to a passive $SF_6$ tracer. We also present laboratory measurements of the infrared spectrum of $SF_6$ and find good agreement with previous studies. We calculate the resulting radiative forcings and efficiencies to be, on average, very similar to those reported previously. Our values for the 20, 100 and 500-year global warming potentials are 18,000, 23,800 and 31,300, respectively.





**1 Introduction**

Sulphur hexafluoride ($SF_6$) is an anthropogenic greenhouse gas which is mainly used as an electrical insulator, with other applications as a quasi-inert gas. Although its main sources are in the Northern Hemisphere, its atmospheric abundance is increasing globally in response to these emissions and its long atmospheric lifetime. $SF_6$ is characterised by large absorption cross-sections for terrestrial infrared radiation such that the presently increasing $SF_6$ abundance will contribute a positive radiative forcing over many centuries. The important known removal sources are electron attachment and photolysis. Recently, we have also measured bimolecular rate constants for the reaction of $SF_6$ with mesospheric metals (Totterdill *et al.*, 2015).

Harnisch and Eisenhauer (1998) reported that $SF_6$ is naturally present in fluorites, and outgassing from these materials leads to a natural background atmospheric abundance of 0.01 pptv. However, at present the anthropogenic emissions of $SF_6$ exceed the natural ones by a factor of 1000 or more and are responsible for the rapid increase in its atmospheric abundance. Surface measurements show that $SF_6$ increased by about 7%/year during the 1980s and 1990s (Geller *et al.*, 1997; Maiss and Brenninkmeijer, 1998).

$SF_6$ provides a useful tracer of atmospheric transport in both the troposphere and stratosphere. Rates for transport of pollutants into, within, and out of the stratosphere are important parameters that regulate stratospheric composition. The basic characteristics of the stratospheric Brewer-Dobson (B-D) circulation are known from observations of trace gases such as $SF_6$: air enters the stratosphere at the tropical tropopause, rises at tropical latitudes, and descends at middle and high latitudes to return to the troposphere. Understanding the rate of this transport on a global scale is crucial in order to predict the response of stratospheric ozone to climatic or chemical change. $SF_6$ is essentially inert in the troposphere to middle stratosphere and is removed by electron attachment and photolysis in the upper stratosphere and mesosphere (Ravishankara *et al.*, 1993). This tracer therefore provides an ideal probe of transport on timescales of importance in the stratospheric circulation and quantitative information on mean air mass age for the lower and middle stratosphere.

The mean age-of-air (AoA) is the interval between the time when the volume mixing ratio of a linearly increasing atmospheric tracer reaches a certain value at a given location in the stratosphere and an earlier time when this mixing ratio was reached at a reference location. Mean AoA is expressed as (Hall and Plumb, 1994; Waugh and Hall, 2002)



$$AoA = t(\chi, l, z) - t(\chi, l_0, z_0) \qquad\qquad (E1)$$


where $t$ is time, $\chi$ is the volume mixing ratio, $l$ and $z$ are latitude and altitude, and the 0 subscripts
denote the reference latitude and altitude which are chosen to be the upper tropical troposphere
(latitude = $1^\circ$N, altitude = 13.9 km). In principle the trend of AoA can be used to diagnose
changes in the strength of Brewer-Dobson circulation (BDC); in practice, however, it is very
difficult to obtain unambiguous results on trends from this or any other trace gas (Garcia *et al.*,
2011). Ideally, AoA should be determined experimentally using a tracer with very small (or
zero) chemical sink in the stratosphere or mesosphere. Otherwise, a correction must be applied
to account for this loss. A correction would also be necessary for any non-linear tropospheric
growth. However, for the period considered for diagnosing age-of-air in this paper (2002-2007)
the growth of $SF_6$ is approximately linear, so we can reasonably neglect such a correction for
$SF_6$-derived AoA (Hall and Plumb, 1994).
Ravishankara *et al.* (1993) reported the atmospheric lifetime of $SF_6$ to be 3200 years by
considering electron attachment and vacuum ultraviolet (VUV) photolysis. They also studied
the loss of $SF_6$ by reaction with $O(^1D)$ but found the rate too slow to be important. They
deduced that electron attachment was the dominant loss process and quantified this process
using a 2-D model, wherein they assumed that all $SF_6$ molecules are destroyed after attachment
of an electron (with a rate constant of $10^{-9}$ $cm^3$ molecule$^{-1}$ s$^{-1}$). They therefore argued that their
lifetime of 3200 years could be a lower limit, but clearly this result depends on the accuracy of
the 2-D electron density, which was calculated using only photochemistry. Morris *et al.* (1995)
subsequently extended the work of Ravishankara *et al.* (1993) by including an ion chemistry
module in the same 2-D model. They also made other assumptions to maximise the impact of
electron attachment on $SF_6$ loss and derived a lifetime as low as 800 years (which could be
further sporadically decreased by large solar proton events). Using a 3-D middle atmosphere
model, Reddmann *et al.* (2001) estimated the lifetime to be 472 years when $SF_6$ is irreversibly
destroyed purely by direct electron attachment and to be 9379 years when $SF_6$ loss is assumed
to occur only via indirect loss (via the formation of $SF_6^-$) and ionization via the reactions with
$O_2^+$ and $N_2^+$. They concluded that the estimated lifetime depends strongly on the electron
attachment mechanism, because the efficiency of this process as a permanent removal process
of $SF_6$ depends on the competition between reaction of $SF_6^-$ with H and HCl, and
photodetachment and reaction with O and $O_3$. Here we extend on the above studies and
investigate the $SF_6$ lifetime using a state-of-the-art 3-D chemistry climate model with a domain





from the surface to 140 km. Our modelled electron density is based on results of a detailed ion
chemistry model and we use a detailed methodology for treating the atmospheric background
electrons, which is based on Troe's formalism (Troe *et al.*, 2007a,b; Viggiano *et al.*, 2007).
In addition to determining the $SF_6$ lifetime, in this study we report new measurements of the
infrared absorption cross-sections for $SF_6$ and input these into a line-by-line radiative transfer
model in order to obtain radiative forcings and efficiencies. These values are then used to
determine more accurate values of global warming potentials (GWPs) based on their cloudy
sky adjusted radiative efficiencies. GWP is the metric used by the World Meteorological
Organisation (WMO) and Intergovernmental Panel on Climate Change (IPCC) to compare the
potency of a greenhouse gas relative to an equivalent emission of $CO_2$ over a set time period.
The definitions of these radiative terms are discussed in detail in our recent publication
Totterdill *et al.* (2016).

## 108   2 Methodology

### 109   2.1 WACCM 3D model

To simulate atmospheric $SF_6$ we have used the Whole Atmosphere Community Climate Model
(WACCM). Here we use WACCM 4 (Marsh *et al.*, 2013), which is part of the NCAR
Community Earth System Model (CESM; Lamarque *et al.*, 2012), configured to have 88
pressure levels from the surface to the lower thermosphere ($5.96 \times 10^{-6}$ Pa, 140 km) and a
horizontal resolution of $1.9^o \times 2.5^o$ (latitude × longitude). The model contains a detailed
treatment of middle atmosphere chemistry including interactive treatments of Na and K (Plane
*et al.*, 2015). We use the specified dynamics (SD) version of the model to allow comparison
with observations (see Garcia *et al.*, (2014) for details). The $SF_6$ surface emission flux and
initial global vertical profiles were taken from a CCMI (Chemistry Climate Model Initiative)
simulation using the same version of SD-WACCM with the same nudging analyses (D.
Kinnison, personal communication, 2013).
Lyman-α photolysis is the only $SF_6$ loss reaction in the standard version of WACCM and in
this work we have added the additional processes given in **Table 1**. The rate constants for the
$SF_6$ + metal reactions have been measured in our laboratory for mesospheric conditions
(Totterdill *et al.*, 2015); here we use the experimental values for the reactions with Na and K.
For the photolysis of $SF_6$ we used the standard WACCM methodology but with the updated
Lyman-α cross section from our laboratory of $1.37 \times 10^{-18}$ cm$^2$ molecule$^{-1}$ (Totterdill *et al.*,
2015). The WACCM Lyman-α flux is taken from Chabrillat and Kockarts (1997).



Electron attachment to SF$_6$ plays a major role in its atmospheric removal and so both
dissociative and non-dissociative attachment are considered in this study. The detailed method
is described in a recent paper (Totterdill *et al.*, 2015) and here only a brief summary is given.
The removal process by the attachment of low energy electrons to SF$_6$ can be described using
Troe's theory (Troe *et al.*, 2007a,b; Viggiano *et al.*, 2007). In the middle and lower mesosphere,
electrons are mostly attached to neutral species in the form of anions. However, above 80 km
the concentration of free electrons increases and the direct electron attachment to SF$_6$ becomes
more likely. This can happen either by associative attachment forming the SF$_6^-$ anion which
can then undergo chemical reactions with H, O, O$_3$ and HCl, or by dissociative attachment
forming the SF$_5^-$ anion fragment. The probability $\beta$ of dissociative attachment when an electron
is captured by SF$_6$ is given by
$$\beta(p,T) = \frac{k_{dis}}{k_{at}+k_{dis}} \tag{E2}$$

where $k_{dis}$ is the rate constant for dissociative attachment and $k_{at}$ is the rate constant for
associative attachment. $\beta$ can be expressed as
$$\beta(p,T) = exp(-4587T + 7.74) \times 10^{\left[4.362-0.582log_{10}(p/Torr)-0.0203\left(log_{10}\left(\frac{p}{Torr}\right)\right)^2/5.26\times10^{-4}\right]}$$

$$\tag{E3}$$

where $T$ is the temperature in K and $p$ is the pressure in Torr (Totterdill *et al.*, 2015).
We include both associative and dissociative electron attachment using WACCM-predicted
electron concentrations (see **Table 1**). Note that the SF$_6^-$ anion is not modelled directly. Instead
the SF$_6$ attachment loss rate is calculated by multiplying $k_{at}$ by the probability of permanent
destruction of the resulting SF$_6^-$ (reactions of SF$_6^-$ with H and HCl) to the sum of these reactions
and processes which recycle SF$_6^-$ to SF$_6$ (reactions with O and O$_3$, and photodetachment)
(Morris *et al.*, 1995).
In order to use a realistic electron concentration, the role of negative ions in the $D$ region must
be considered. Therefore, a scaling factor was introduced that converts the standard WACCM
electron concentrations, which are calculated from charge balance with the five major positive
$E$ region ions (N$^+$, N$_2^+$, O$^+$, O$_2^+$ and NO$^+$), to more realistic electron concentrations. We have
recently incorporated the Sodankylä Ion Chemistry (SIC) model into the standard version of
WACCM to produce a new version (WACCM-SIC) containing a detailed $D$ region ion





chemistry with cluster ions and negative ions (Kovács *et al.*, 2016). The mesospheric positive
and negative ions in WACCM-SIC are listed in **Table 2**. The electron scaling factor in each
grid box of WACCM was then defined as the annually averaged ratio of $[e]_{WACCM-SIC}/[e]_{WACCM}$
for the year 2013, where $[e]_{WACCM-SIC}$ is the electron density calculated from WACCM-SIC
and $[e]_{WACCM}$ from the standard WACCM.
The scaling factor, which varies with altitude and latitude, is shown in **Figure 1** (bottom panel)
together with the electron densities from the standard WACCM (top panel) and WACCM-SIC
(middle panel) models. The annually averaged electron concentration in the WACCM-SIC
model is significantly smaller in the lower and middle mesosphere than in the standard
WACCM, which is expected because of negative ion formation. Note that in the upper
mesosphere (70 - 80 km) the electron density in WACCM-SIC is larger than WACCM. This
results from the inclusion of medium energy electrons (MEE) (electrons with energy between
30 keV and 2MeV) in WACCM-SIC. **Figure 2** shows the effect of MEE by comparing
WACCM-SIC runs with and without this source of ionization in the upper mesosphere
included. To describe the effect of ionization, WACCM-SIC uses ionization rates (*I*) as a
function of time and pressure which were calculated from the spectra based on the proton
energy-range measurements in standard air as described by Verronen *et al.* (2005). According
to Figure 3 of Meredith *et al.* (2015), the annually averaged medium energy electron flux for
2013 approximately corresponds to the long-term, 20-year average. This allows us to assume
that the annually averaged electron density of 2013 from WACCM-SIC can be used to scale
the long-term simulations using the standard WACCM aimed at determining the atmospheric
lifetime of $SF_6$.
The WACCM simulation included five different $SF_6$ tracers in order to quantify the importance
of different loss processes. All of these tracers used the same emissions but differed in their
treatment of $SF_6$ loss reactions. One $SF_6$ tracer included no atmospheric loss (i.e. a passive
tracer). Three tracers included one of the following loss processes for $SF_6$: (i) reaction with
mesospheric metals (Na, K), (ii) electron attachment, and (iii) UV photolysis. Finally, one 'total
reactive' $SF_6$ tracer included all three loss processes. This total reactive tracer should be the
most realistic and was used in the radiative forcing calculations. WACCM was run for the
period 1990-2007, and the first five years were treated as spin-up. For the analysis the monthly
mean model outputs were saved and later globally averaged for the lifetime calculations.



### 2.2 Infrared absorption spectrum and radiative forcing

Previous quantitative infrared absorption spectra of $SF_6$ have been compared in Hodnebrog *et al.* (2013) (their Table 12). There are differences of ~10% between existing integrated cross-section estimates, and the measurements cover different spectral ranges. We therefore performed a more complete set of measurements over a wider spectral range, in order to reduce uncertainty in the absorption spectrum and hence the radiative efficiency of $SF_6$. Measurements were taken using an experimental configuration consisting of a Bruker Fourier transform spectrometer (Model IFS/ 66), which was fitted with a mid-infrared (MIR) source used to generate radiation which passed through an evacuable gas cell with optical path length 15.9 cm. The cell was fitted with KBr windows, which allow excellent transmission between 400 and 40,000 cm$^{-1}$. The choice of source and window were selected so as to admit radiation across the mid IR range where bands of interest are known to occur. Room temperature (296 ± 2 K) measurements were carried out between 400 and 2000 cm$^{-1}$ at a spectral resolution of 0.1 cm$^{-1}$ and compiled from the averaged total of 128 scans to 32 background scans at a scanner velocity of 1.6 kHz. Gas mixtures were made using between 8 and 675 Torr of $SF_6$ diluted up to an atmosphere using $N_2$, according to the method described in Totterdill *et al.* (2016).

Radiative forcing calculations were made using the Reference Forward Model (RFM) (Dudhia, 2013) which is a line-by-line radiative transfer model based on the previous GENLN2 model (Edwards, 1987). Results obtained from this model were validated against the DISORT radiative transfer solver (Stamnes *et al.*, 2000) included within the libRadtran (Library for Radiative Transfer) package (Mayer and Kylling, 2005). A full description of these models and parameters used alongside discussion of treatment of clouds and model comparison is also given in Totterdill *et al.* (2016).

### 3 Results

### 3.1 Global distributions of SF6 from WACCM simulations

**Figure 3** shows typical zonal mean profiles of the WACCM $SF_6$ tracers in the north and south polar regions for different seasons, compared to MIPAS observations for the year 2007 (Haenel *et al.*, 2015). Although the MIPAS $SF_6$ data provides much more coverage horizontally and vertically compared to in situ aircraft and balloon data, it has only been validated up to 35 km (Stiller *et al.*, 2008). At higher altitudes validation is not possible due to the lack of suitable reference data. Details of the validation of the MIPAS data version used here (V5h_SF6_20 for the full resolution product from 2004 and earlier; V5r_SF6_222 and V5r_SF6_223 for the



reduced resolution period of 2005 and later) can be found in Haenel *et al.* (2015), including
Figure S-2 of their supplementary material. The WACCM passive $SF_6$ tracer has a mixing ratio
profile that is fairly constant with altitude until around 70 km, after which it decreases.
Comparison of the tracers that include loss processes show that removal of $SF_6$ is dominated
by electron attachment, with a small contribution direct from photolysis. The mesospheric
metals make a negligible contribution because the Na and K layers occur in the upper
mesosphere above 80 km (with peaks around 90 km), and the concentrations of these metal
atoms are too low. As is clear from **Figure 3**, the model simulation and satellite observations
agree within the atmospheric variability, which becomes relatively large above 30 km
especially at high latitudes, although the model is systematically larger than the observations
above 20 km. The time variation of modelled $SF_6$ shown in **Figure 4** corresponds to an
emission rate (slope) of $6.5 \times 10^{-3}$ Tg/year, i.e. a 0.29 pptv/year increase in global mean volume
mixing ratio, and a volume mixing ratio of 6.4 pptv by the end of 2007.
**Figure 5** shows the zonal mean annual mean $SF_6$ distribution from the five WACCM tracers
and MIPAS observations for 2007. **Figure 5a** (and **Figure 3**) shows that there is a rapid
decrease in $SF_6$ above 75 km even for the inert tracer. This can be explained by diffusive
separation, which becomes pronounced in the upper mesosphere because $SF_6$ is a relatively
heavy molecule compared to the mean molecular mass of air molecules (cf. Garcia *et al*. (2014),
where similar behaviour is seen for $CO_2$, another relatively heavy molecule). Panels (a)-(c) of
the figure all show $SF_6$ decreasing above ~80 km, and panels (a) and (c) are almost identical,
while in panel (b) the decrease begins a little lower. This is all consistent with the notion that
metals do not affect $SF_6$ and photolysis contributes only slightly. The fact that diffusive
separation prevents $SF_6$ from reaching altitudes where photolysis is faster must be contributing
to the very long lifetime found when photolysis is the only loss considered. By contrast, in
**Figure 5d** $SF_6$ decreases rapidly above 70 km, which is related to the fact that loss via electron
attachment is important at these lower altitudes. Thus, in this case, $SF_6$ loss occurs below the
altitudes where diffusive separation is important (and where air density is higher), which makes
it a much more effective loss mechanism. The WACCM $SF_6$ tracer that includes all loss
processes (**Figure 5e**) has a very similar distribution to that which only treats loss due to
electron attachment (**Figure 5d**), which emphasises how this process dominates $SF_6$ loss in the
model. This model tracer can be compared to the MIPAS observations in **Figure 5f**, which
shows that WACCM agrees reasonably well with the measurements in the lower stratosphere
(note the smaller altitude range in panels (e) and (f) of **Figure 5**). Finally, it is also clear that



WACCM $SF_6$, even with all losses considered, decreases with altitude much more slowly at all
latitudes than MIPAS $SF_6$. This could indicate a problem with the model's meridional transport.
However, a too-fast BDC would tend to produce low levels of $SF_6$ at middle and high latitudes
in the descending branch, which does not seem to be the case. Therefore, at least two other
possible scenarios could be responsible for the discrepancy: $SF_6$ loss in WACCM is still
somewhat underestimated despite the inclusion of the electron attachment, or MIPAS $SF_6$ is
biased low above ~20 km.
**3.2 Atmospheric lifetime**
The atmospheric lifetime is defined as the ratio of the atmospheric burden to the atmospheric
loss rate. This definition was used to calculate annual mean lifetime values from the WACCM
output containing the individual rates for the different loss processes. During the simulation
the total atmospheric burden of $SF_6$ increased linearly as expected (see **Figure 4**) from $3.4 \times 10^{32}$
molecules with an annual increment of $2.3 \times 10^{31}$ molecules/year. **Figure 6** shows the variation
in $SF_6$ lifetime from 1995 to 2007, corresponding to a full solar cycle (the solar minima
occurred in May 1996 and January 2008). The figure demonstrates that the lifetime has a strong
dependence on solar activity, being anti-correlated with the solar radio flux at 10.7 cm (2800
MHz) (Tapping, 2013) which ranges over $(72 - 183) \times 10^{-22}$ W m$^{-2}$ Hz$^{-1}$, with an average value
of $90.3 \times 10^{-22}$ W m$^{-2}$ Hz$^{-1}$. The mean $SF_6$ lifetime and range over the same solar cycle period $\tau$
= 1278 years, with a range from 1120 to 1475 years. The annual averaged electron number
density in the polar regions is also plotted in **Figure 6**; as expected, it is correlated with the
10.7 cm radio emission (Tapping, 2013).
As noted in the Introduction, the $SF_6$ lifetime has been reported to be 3200 years by
Ravishankara *et al.*, (1993). For this they used a total electron attachment rate constant of $k_{EA}$
$= 10^{-9}$ cm$^3$ s$^{-1}$. In Morris *et al.* (1995) the calculated lifetime decreased to 800 years by
considering ion chemistry and assuming that the associative attachment forming $SF_6^-$ does not
regenerate the parent molecule, thereby obtaining a lower limit for the lifetime. Reddmann *et*
*al.* (2001) estimated the lifetime to be 472 yr when $SF_6$ is irreversibly destroyed purely by
direct electron attachment and to be 9379 yr when $SF_6$ loss is assumed to occur only via indirect
loss (via the formation of $SF_6^-$) and ionization via the reactions with $O_2^+$ and $N_2^+$. In the present
study we have directly applied Troe's theory (Troe *et al.*, 2007a,b; Viggiano *et al.*, 2007) to
determine the efficiency of electron attachment as a function of temperature and pressure, and



the branching ratio for dissociative attachment (equation E2), which we extrapolated to
mesospheric conditions (Totterdill *et al.*, 2015).
Our estimated *partial* lifetime of SF$_6$ due to photolysis for the SF$_6$ tracer which includes all
loss processes is 48,000 yr, which is considerably longer than that the 13,000 yr determined by
Ravishankara *et al.* (1993) despite our Lyman-α cross section ($1.37 \times 10^{-18}$ cm$^2$, **Table 1**) being
only ~22% smaller than the value the value measured by Ravishankara *et al.* ($1.76 \times 10^{-18}$ cm$^2$).
One reason why our photolysis-related partial lifetime is longer is that WACCM includes
diffusive separation, which was not described in the earlier 2-D model study. The inclusion of
diffusive separation reduces sharply the abundance of SF$_6$ at high altitudes, where photolysis
is most effective. Another contributing factor could be that the VUV photolysis is important
only above 80 km, while in our model runs SF$_6$ is mostly destroyed by electron attachment,
which results in less being transported into this upper mesospheric region. When we analyse
our WACCM SF$_6$ tracer which is subject to photolysis loss only, the resulting steady-state
*overall* lifetime for the last model year (2007) is 17,200 yr which is only 32% larger than the
value of Ravishankara *et al.* (1993) and thus more consistent with the difference in the Lyman–
α cross sections. Finally, if we do not include the electron scaling factor to reduce the electron
density below 80 km due to negative ion formation, then the SF$_6$ lifetime decreases to 776 years
(not shown), which is similar to the value obtained by Morris *et al*. (1995).
**3.3 Impact of SF$_6$ loss on mean age of stratospheric air**
As SF$_6$ is a chemically stable molecule in the stratosphere and troposphere, and has an almost
linearly increasing tropospheric abundance, its atmospheric mixing ratio is often used to
determine the mean age of stratospheric air. This is an important metric in atmospheric science
as the distribution of ozone and other greenhouse gases depends significantly on the transport
of air into, within, and out of the stratosphere. WACCM contains an idealized, linearly-
increasing age-of-air tracer (AOA1) that provides model age values for model experiments
(Garcia *et al.*, 2011).
Age-of-air has generally been derived from observations by treating SF$_6$ as a passive (non-
reactive) tracer. The assumption is that the global loss rate is too slow to significantly affect
the lifetime. This was confirmed by Garcia *et al.* (2011) when only photolysis was included.
However, when loss via electron attachment is also considered, the lifetime may become short
enough that this assumption is no longer valid, in which case the stratospheric mixing ratio
would appear to correspond to an earlier tropospheric mixing ratio than in reality. We have



compared the passive WACCM $SF_6$ tracer with that subject to all loss processes, which yields
the new lifetime of 1278 yr. The difference between these two tracers indicates the error in the
derived age-of-air that would arise in the real atmosphere if $SF_6$ is assumed to be a passive
tracer. The error caused by chemical removal can be expressed as:
$$\Delta(AoA) = AoA(\text{reactive tracer}) - AoA(\text{passive tracer}) \qquad (E4)$$
where $\Delta(AoA)$ is the difference in the age-of-air value caused by chemical loss, AoA(reactive
tracer) is the calculated age-of-air considering the chemical removal, and AoA(passive tracer)
is the value obtained from a non-reactive tracer. The expression for the age-of-air at any point
in the stratosphere can be obtained from a simplified version of (E1) that is derived from a
Taylor series expansion, retaining only the linear term; then it is expressed as
$$AoA = [(\chi_0(SF_6) - \chi(SF_6))/ \, r(SF_6)] \qquad (E5)$$
where $\chi(SF_6)$ and $\chi_0(SF_6)$ are the $SF_6$ volume mixing ratios at the actual and the reference
(tropical tropopause) points, respectively, while $r(SF_6)$ is the rate of increase of tropospheric
$SF_6$. In our simulations $r(SF_6)$ is 0.29 pptv/year (**Figure 4**), which is an approximation as the
growth rate is not constant in reality. Stiller *et al*. (2012) report a value of 0.24 pptv/year based
on observations. These two simplifications will lead to deviations between WACCM and
MIPAS age data. If (E5) is substituted into (E4) then the error in age-of-air will be:
$$\Delta(AoA) = (\chi(SF_{6,\,passive}) - \chi(SF_{6,\,reactive}))/ \, r(SF_6) \qquad (E6)$$
This error, along with the mean age itself, was calculated from WACCM output for 2007.
**Figure 7** shows the annual mean ages determined from the WACCM simulation from 2002-
2007 using the total reactive and the inert $SF_6$ tracers and the idealized AOA1 age tracer. There
is a clear difference between the age values derived from the passive $SF_6$ and the idealized AoA
tracer. If equation (E5) is used to determine the age values there is no guarantee that the age
values derived from the two tracers will be identical; the rate was determined from the increase
of the $SF_6$ burden (0.29 pptv/year) and this was provided by the linear fit (**Figure 4**), which
can misrepresent the growth rate at any time. **Figure 7** also shows the difference between the
age values obtained from the reactive and inert $SF_6$ tracers. It can be seen that consideration of
the reactive $SF_6$ tracer does indeed affect the determined mean age values, mostly where
electron attachment dominates. The age estimates at high latitudes are most sensitive to
chemical loss because the air that reaches these locations has descended from the high altitudes



where $SF_6$ loss predominantly occurs. According to the MIPAS satellite observations (Stiller
*et al.*, 2012; Haenel *et al.*, 2015.), the derived age value over the tropical lower stratosphere at
25 km is slightly more than 3 years, while the WACCM simulations with the reactive $SF_6$ tracer
predicts 3 years. Comparing **Figures 7a** and **7b**, the effect of chemical removal in this region
is minor (0.01 year or 0.5% change) and therefore it does not have much impact on the inferred
atmospheric transport. At the poles the effect is much more significant; the difference at 25 km
between the reactive and inert $SF_6$ tracers is up to 0.55 years (9%). This means that in the
troposphere-stratosphere low latitude regions the effect of chemical removal is not very
significant and the error on the estimated mean age caused by the assumption of $SF_6$ being a
passive tracer is not important. However, the effect of chemical removal becomes more
significant at high latitudes.
We can also compare modelled and observed mean age values in the lower stratosphere (20
km). **Figure 8** shows the mean age profiles from WACCM tracers, ER-2 observations (Hall et
al., 2009) and our analysis of MIPAS $SF_6$ satellite data at 20 km. From this it can be seen that
in the tropical region the mean age values are similar between the idealized age tracer and the
inert and reactive $SF_6$ tracers. This is consistent with no loss of $SF_6$ having occurred in air
parcels in the deep tropics. At high latitudes there is up to 0.5 year difference in the modelled
mean ages, with the reactive $SF_6$ tracer producing the oldest apparent age. The differences in
mean age between the tracers is larger in the SH polar region than in the NH because the polar
region is less well mixed. The tendency is very similar when we compare the WACCM mean
ages to the MIPAS observations. Note that the satellite observations show more seasonal
variability in the middle and high latitudes than in the tropics.
**3.4 Radiative Efficiency and Forcing**
To determine the radiative efficiency and global warming potential of $SF_6$, integrated cross-
sections were taken from the GEISA: 2011 Spectroscopic Database (Varanasi, 2011), the
HITRAN 2012 Molecular Spectroscopic Database (Rothman *et al*., 2012), and were also
measured in this study. The literature values are presented in **Table 4** for comparison with our
experimentally determined values and the full $SF_6$ spectrum obtained in this study is given in
**Figure 9**. In our study the spectrometer error is $\pm1.0\%$ for all experiments, and the uncertainty
in the sample concentrations of $SF_6$ was calculated to be 0.7%. Spectral noise was averaged at
$\pm5\times10^{-21}$ $cm^2$ molecule$^{-1}$ per 1 cm$^{-1}$ band. However, at wavenumbers <550 cm$^{-1}$, towards the
edge of the mid infrared where opacity of the KBr optics increases, this value was $1\times10^{-20}$ $cm^2$





molecule$^{-1}$ per 1 cm$^{-1}$ band. The error from determining the scaling cross-section was 5%. This
results in an average overall error of ±5% in the cross-sections.
The intensities of the main SF$_6$ absorption bands (925-955 cm$^{-1}$) measured in this study are 7%
greater than those reported by (Hurley, 2003), 1% greater than GEISA (Varanasi, 2011) and 1%
lower than those given in HITRAN (Rothman *et al.*, 2012) (**Table 4**). Comparison of our results
against Varanasi (2011) between 650 and 2000 cm$^{-1}$ gives an agreement within 9%. Note that
these differences are within the combined error of both experiments.
The instantaneous and stratospheric adjusted SF$_6$ radiative efficiencies in clear and cloudy sky
conditions are given in **Table 5**. These are also presented as present-day radiative forcings
employing a current surface concentration of 9.3 pptv (NOAA, 2016) (see **Figure 4**). The
radiative efficiency was calculated in the RFM for each month between 90ºS and 90ºN at
latitudinal resolutions (on which the data was averaged to obtain the zonal mean vertical
profile) of 1.5º and 9.0º. The tropopause used the standard WMO lapse rate definition (see
Totterdill *et al.*, 2016). **Figure 10** shows the seasonal-latitudinal variation of the instantaneous
clear sky radiative forcing for SF$_6$ on the high (1.5º) and low (9º) resolution grids. Employing
profiles averaged over the lower resolution grid gives an average forcing within 1% of the
higher resolution grid. Using only a single annually averaged global mean profile led to a 10%
error in radiative forcing when compared to our monthly resolved high resolution profile,
supporting the findings of Freckleton *et al.* (1998) and Totterdill *et al.* (2016).
A selection of experiments were carried out over a range of months and latitudes to investigate
the sensitivity of the forcing calculations to the bands used. The average contributions from the
main bands were compared against the calculation with the full measured spectrum. The results
showed that the 580 – 640 and 925 – 955 cm$^{-1}$ bands contribute almost 99% to the instantaneous
radiative forcing. Our forcing calculations suggest that the SF$_6$ minor bands contribute only a
small amount to the final value. This means that deviations between our experimentally
determined spectra and those in the literature only result in a significant change to previously
published radiative forcings and efficiencies when that deviation occurs over a major band.
The SF$_6$ adjusted cloudy sky radiative efficiency published by the IPCC AR5 report and used
to determine its GWP values is 0.57 Wm$^{-2}$ ppbv$^{-1}$ (Myhre *et al.*, 2013). This compares to the
adjusted cloudy sky radiative efficiency determined in this study of 0.59 Wm$^{-2}$ ppbv$^{-1}$. A review
on radiative efficiencies and global warming potentials by Hodnebrog *et al.* (2013) provides a





comprehensive list of all published values for these parameters for many species including $SF_6$.
They established the range of published radiative efficiencies for $SF_6$ to be $0.59 – 0.68$ $Wm^{-2}$
$ppbv^{-1}$, with a mean value of $0.56$ $Wm^{-2}$ $ppbv^{-1}$. They also made their own revised estimate
using an average of the HITRAN (Rothman *et al.*, 2012) and GEISA (Hurley, 2003; Varanasi,
2011) spectral databases and found a best estimate of $(0.565 \pm 0.025)$ $Wm^{-2}$ $ppbv^{-1}$. Their mean
value for radiative efficiency is very close to that determined in this study using similar
conditions ($0.59$ $Wm^{-2}$ $ppbv^{-1}$).

**3.5 Global Warming Potential**

**Table 6** gives our estimates of the 20, 100 and 500-year GWPs based on cloudy sky adjusted
radiative efficiencies of $SF_6$ compared with IPCC AR5 values (IPCC, 2013). Our 20, 100 and
500-year global warming potentials for $SF_6$ are 18,000, 23,800 and 31,300 respectively. The
20-year and 100-year values are 3% greater and 1% greater, respectively, than their IPCC
counterparts and the 500-year GWP is 4% smaller than its AR4 counterpart (Forster *et al.*,
2007). Forcing efficiencies determined in this study are somewhat higher than previously
published values, which imply a higher value for GWP. However, our shorter atmospheric
lifetimes would lead to a smaller GWP estimate. The trade-off between these competing effects
is apparent in **Table 6**, where $SF_6$ exhibits a 20-year GWP that is slightly larger than the IPCC
value, while the 500-year GWP is slightly smaller. The radiative efficiency effect is most
obvious for the case of the 20-year GWP where, because the atmospheric lifetime of $SF_6$ is
1278 years, the species does not have time for any significant loss to occur.

**4 Conclusions**

The 3D Whole Atmosphere Community Climate Model was used to simulate the $SF_6$
atmospheric distribution over the period of 1995-2007. From the concentrations and the
knowledge of the electron attachment, photolysis and metal reaction rates we determined the
atmospheric lifetime which shows a significant dependence on the solar cycle due to varying
electron density. The mean $SF_6$ atmospheric lifetime and $1\sigma$ variation over a solar cycle were
determined to be 1278 years (ranging from 1120 to 1475 years), which is different to previously
reported literature values and much shorter than the widely quoted value of 3200 years. The
reason is our more detailed treatment of electron attachment using a new formalism to describe
both associative and dissociative attachment, and the use of a detailed model of *D* region ion
chemistry to evaluate the partitioning of electrons and negative ions below 80 km.





Based on this new estimate of the $SF_6$ lifetime, we find that the derived mean age of
stratospheric air from observations can be slightly affected by the atmospheric removal of $SF_6$.
In the polar region the age-of-air values differ by up to 9% when the values from inert and
reactive model tracers are compared, suggesting that $SF_6$ loss does not have a large influence
on the age values but that it should be included in detailed analyses.
We also re-investigated the radiative efficiency and global warming potential of $SF_6$. Our
radiative efficiency value reported here, $0.59 \pm 0.045$ $Wm^{-2}$ $ppbv^{-1}$, is slightly higher than the
IPCC AR5 estimate of $0.57$ $Wm^{-2}$ $ppbv^{-1}$. The global warming potentials of $SF_6$ for 20, 100
and 500 years have been determined to be 18,000, 23,800 and 31,300, respectively. We find
that our revised lifetime and efficiency values somewhat cancel each other out so overall do
not play a significant role in modifying the GWP estimates.
**Acknowledgements**
This work was part of the MAPLE project funded by research grant NE/J008621/1 from the
UK Natural Environment Research Council, which also provided a studentship for AT. The
authors are also thankful to Prof Jürgen Troe for the helpful discussions related to the electron
attachment to $SF_6$.



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





**Tables**
**Table 1**. SF$_6$ loss reactions included in WACCM.

| Loss process | Rate constant | Reference and comments |
|---|---|---|
| Na + SF$_6$ | $k = 1.80 \times 10^{-11} \exp(-590.5/T)$ | From Totterdill *et al.*, (2015) Refitted for mesospheric temperatures 215-300K. |
| K + SF$_6$ | $k = 13.4 \times 10^{-11} \exp(-860.6/T)$ | From Totterdill *et al.*, (2015) Refitted for mesospheric temperatures 215-300K. |
| Electron attachment | Associative attachment: $k_{EA,ass} = k_{at} \times (k_{(SF6- + H)}[H] + k_{(SF6- + HCl)}[HCl]) / (j_{PD} + k_{(SF6- + H)}[H] + k_{(SF6- + HCl)}[HCl] + k_{(SF6- + O3)}[O_3] + k_{(SF6- + O)}[O])$ Dissociative attachment: $k_{EA,diss} = k_{at} \times \beta$, where $\beta$ is the fraction of SF$_6^-$ that dissociates into SF$_5^-$. | Totterdill *et al.*, (2015). |
| Photolysis | Lyman-α: $\sigma(121.6 \text{ nm}) = 1.37 \times 10^{-18} \text{ cm}^2$ Parameterised expression over the range of 115-180 nm, based on previous measurements. | Totterdill *et al.*, (2015). |






**Table 2.** Positive and negative ions included in WACCM-SIC.

| | |
|---|---|
| Positive ions | $O_2^+$, $O_4^+$, $NO^+$, $NO^+(H_2O)$, $O_2^+(H_2O)$, $H^+(H_2O)$, $H^+(H_2O)_2$, $H^+(H_2O)_3$, $H^+(H_2O)_4$, $H^+(H_2O)_5$, $H^+(H_2O)_6$, $H_3O^+(H_2O)_2(CO_2)$, $H_3O^+(OH)$, $O_2^+(CO_2)$, $H_3O^+(OH)(CO_2)$, $H_3O^+(OH)(H_2O)$, $O_2^+(H_2O)(CO_2)$, $O_2^+(H_2O)_2$, $O_2^+(N_2)$, $NO^+(H_2O)_2$, $H^+(H_2O)(CO_2)$, $O^+$, $N^+$, $N_2^+$, $NO^+(H_2O)_3$, $O_4^+$, $H^+(H_2O)_2(CO_2)$, $H^+(H_2O)_2(N_2)$ |
| Negative ions | $O_3^-$, $O^-$, $O_2^-$, $OH^-$, $O_2^-(H_2O)$, $O_2^-(H_2O)_2$, $O_4^-$, $CO_3^-$, $CO_3^-(H_2O)$, $CO_4^-$, $HCO_3^-$, $NO_2^-$, $NO_3^-$, $NO_3^-(H_2O)$, $NO_3^-(H_2O)_2$, $NO_3^-(HNO_3)$, $NO_3^-(HNO_3)_2$, $Cl^-$, $ClO^-$, $NO_2^-(H_2O)$, $Cl^-(H_2O)$, $Cl^-(CO_2)$, $Cl^-(HCl)$ |


**Table 3.** Partial (reactions with electrons, photolysis, and metals (K, Na)) and total atmospheric
lifetimes (years) of $SF_6$ from different studies. Numbers in parentheses show relative
percentage contribution of loss due to the different processes.

| Study | Lifetime / years | | | |
|---|---|---|---|---|
| | Photolysis | Electron attachment | Total | Model dimensions |
| Ravishankara *et al.* (1993) | 13,000 (24%) | 4210 (76%) | 3200 | 2D |
| Morris *et al.* (1995) | N/A | N/A | 800 | 2D |
| This work | 48,000 (2.6%) | 1339 (97.4%) | 1278 | 3D |




**Table 4.** Integrated absorption cross-sections for $SF_6$ measured in this work and ratios with
values obtained by Hurley (2003), Varanasi (2001) and HITRAN (Rothman *et al.*, 2012).

| Band limits (cm$^{-1}$) | Integrated cross-section (10$^{-16}$ cm$^2$ molec$^{-1}$ cm$^{-1}$) | Ratio of integrated cross-sections in this work to previous studies | | |
|---|---|---|---|---|
| | | Hurley (2003) | Varanasi (2001) | HITRAN |
| 925 - 955 | 2.02 | 1.07 | 1.01 | 0.99 |
| 650 - 2000 | 2.40 | - | 1.09 | - |


**Table 5**. Calculated instantaneous and stratospheric adjusted radiative forcings and radiative
efficiencies of $SF_6$ in clear and all-sky conditions[a].

| | Instantaneous | | Stratospheric adjusted | |
|---|---|---|---|---|
| | Clear | All-sky | Clear | All-sky |
| Radiative forcing (mWm$^{-2}$) | 76.43 | 48.91 | 81.81 | 56.01 |
| Radiative efficiency (Wm$^{-2}$ ppbv$^{-1}$) | 0.77 | 0.50 | 0.85 | 0.59 |

a. Based on present day atmospheric $SF_6$ surface concentration of 9.3 pptv.
**Table 6**. Comparison of 20, 100 and 500-year global warming potentials for $SF_6$ from this work
with values from IPCC (2013).

| | Global Warming Potential | | |
|---|---|---|---|
| | GWP$_{20}$ | GWP$_{100}$ | GWP$_{500}$ |
| This work[a] | 18000 | 23700 | 31300 |
| IPCC (2013)[b] | 17500 | 23500 | 32600[c] |
| Difference (%) (This work – IPCC) | +3% | +1% | -4% |

[a] Based on our atmospheric lifetime of 1278 yrs and RE of 0.59 Wm$^{-2}$ ppbv$^{-1}$.
[b] Based on an atmospheric lifetime of 3200 yrs and RE of 0.57 Wm$^{-2}$ ppbv$^{-1}$ .
[c] Based on an atmospheric lifetime of 3200 yrs and RE of 0.52 Wm$^{-2}$ ppbv$^{-1}$ from IPCC AR4
(Forster *et al.*, 2007).




**Figures**

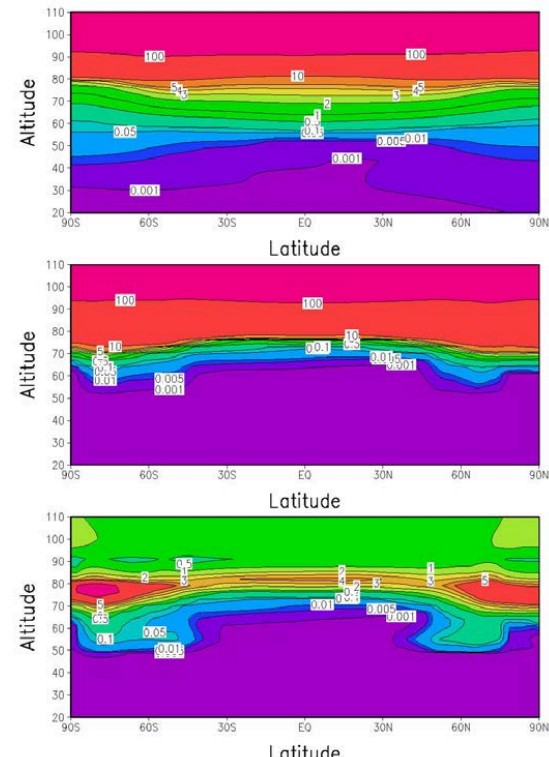


**Figure 1.** Top: annual average electron concentration for 2013 from the standard WACCM
model (in $10^2$ electrons cm$^{-3}$). Middle: annual average electron concentration for 2013 from
WACCM-SIC model (in $10^2$ electrons cm$^{-3}$). Bottom: annually averaged electron scaling factor
for 2013.




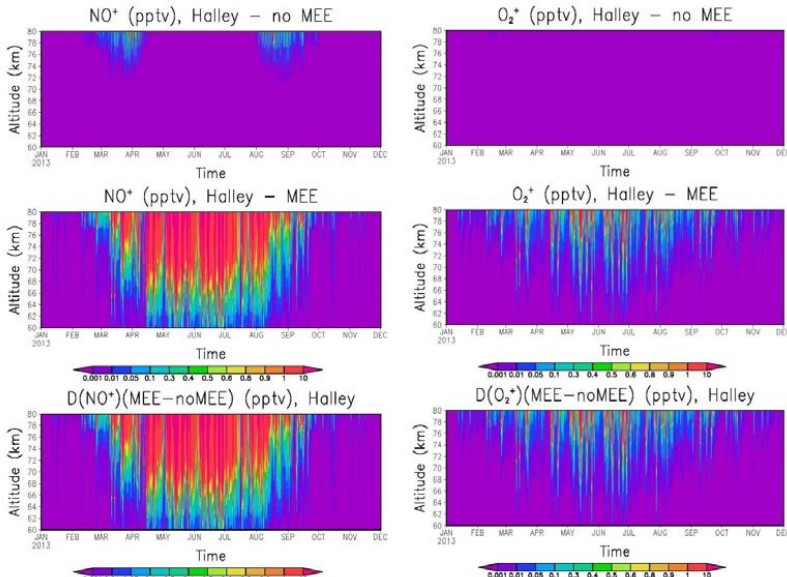


**Figure 2.** Time series of volume mixing ratio profiles (pptv) of $NO^+$ (left panels) and $O_2^+$ (right
panels) above Halley (76°S) from two WACCM-SIC simulations. Top panels show the values
obtained from the model run without medium energy electrons; the middle panels show the run
with medium energy electrons; and the bottom panels show the absolute differences between
the two model runs.




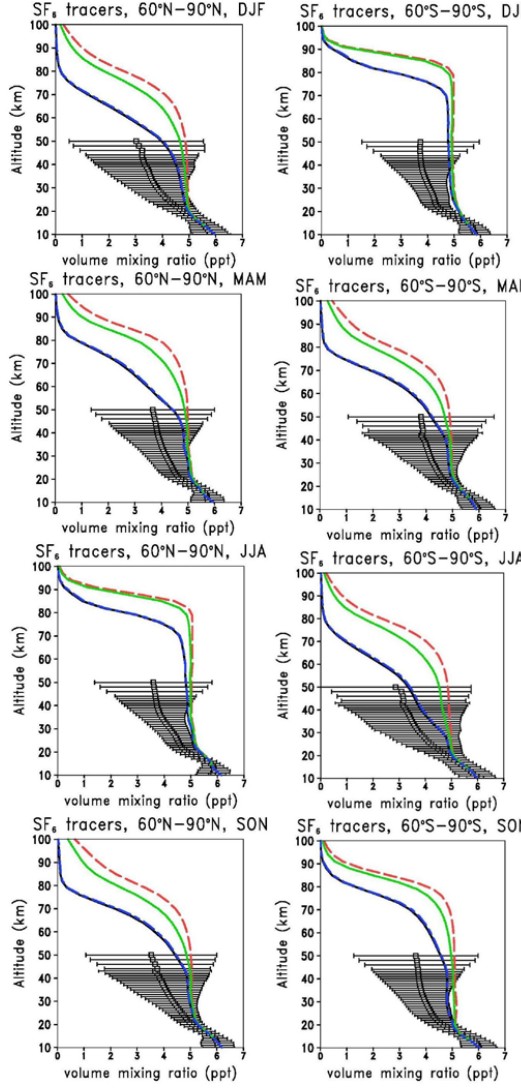


**Figure 3.** Annual volume mixing ratios (pptv) of the different $SF_6$ tracers for the polar regions
(60°N – 90°N and 60°S – 90°S latitudes) in 2007 as a function of altitude for MIPAS satellite
observed $SF_6$ (black symbols with standard deviations for ±1σ) (Stiller *et al.*, 2012), the total
WACCM-$SF_6$ (blue solid line), the photolysis WACCM-$SF_6$ tracer (green solid line) and the
inert WACCM $SF_6$ tracer (red dashed line).






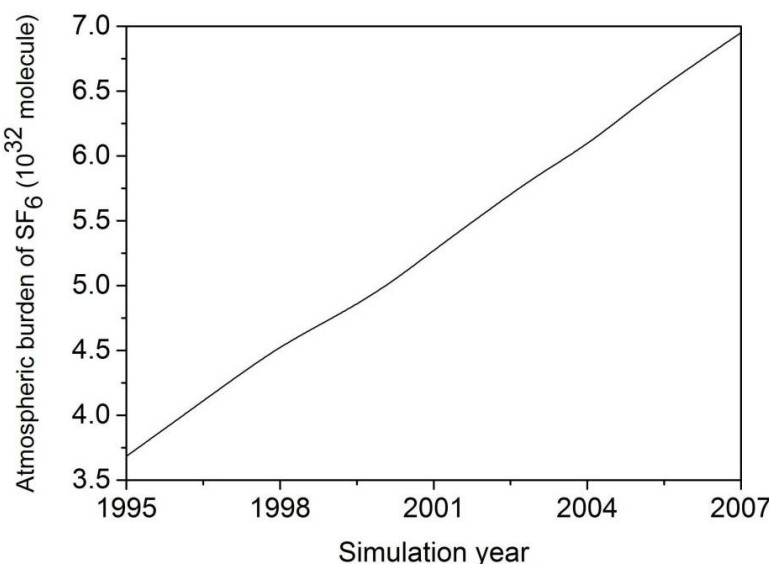


**Figure 4.** Variation of the total annual atmospheric burden of $SF_6$ during the simulation from
1995 to 2007. According to this the emission rate (slope) was determined to be $6.5 \times 10^{-3}$
Tg/year, corresponding to 0.29 pptv/year.





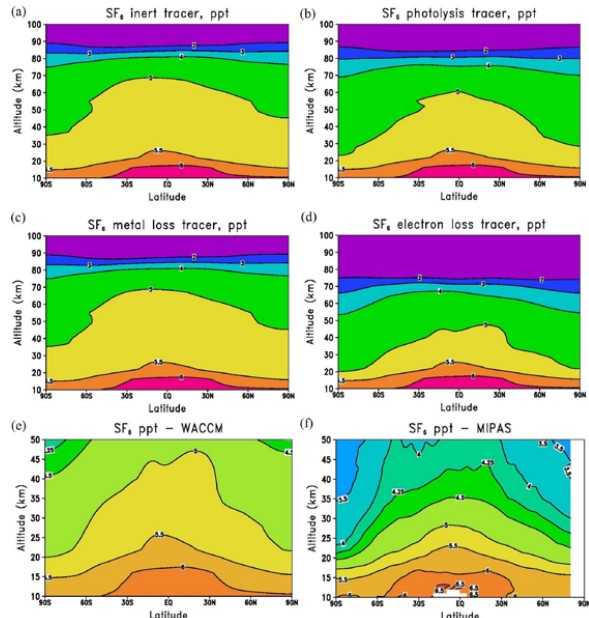


**Figure 5.** Annual zonal mean latitude-height volume mixing ratios (pptv) of the different WACCM SF$_6$ tracers in 2007: (a) inert SF$_6$ tracer; (b) SF$_6$ tracer removed by photolysis only; (c) SF$_6$ tracer removed by mesospheric metals only; (d) SF$_6$ tracer removed by electron attachment only; and (e) total reactive SF$_6$. Panel (f) shows the SF$_6$ volume mixing ratio for 2007 from MIPAS observations. Note the different altitude ranges for panels (a)-(d) and (e)-(f).



629

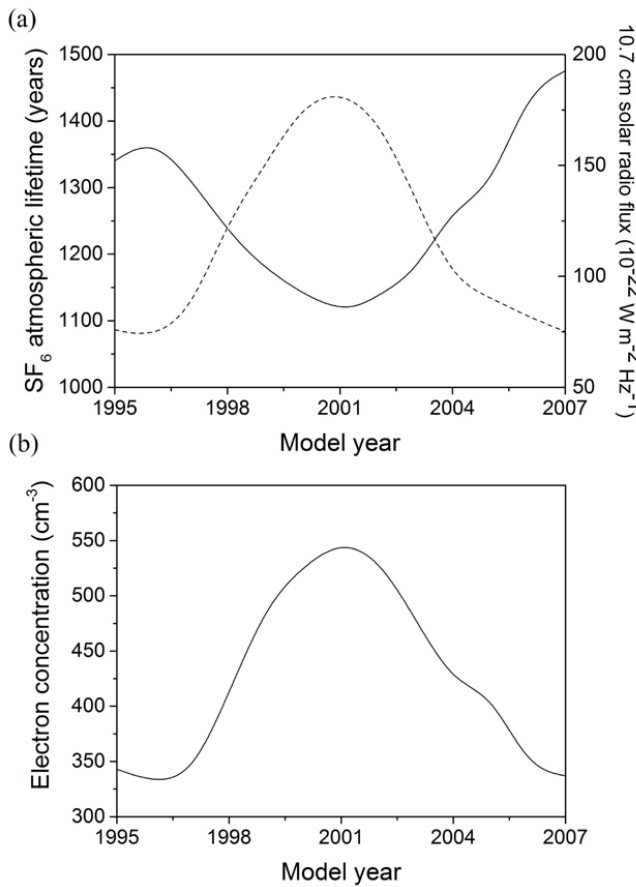

630

**Figure 6.** (a) Variation in atmospheric lifetime of $SF_6$ (solid line) and 10.7 cm solar radio flux
(dashed line) during the WACCM simulation. (b) Variation of the WACCM electron
concentration ($cm^{-3}$) at 80 km, averaged over polar latitudes (60°N – 90°N and 60°S – 90°S).





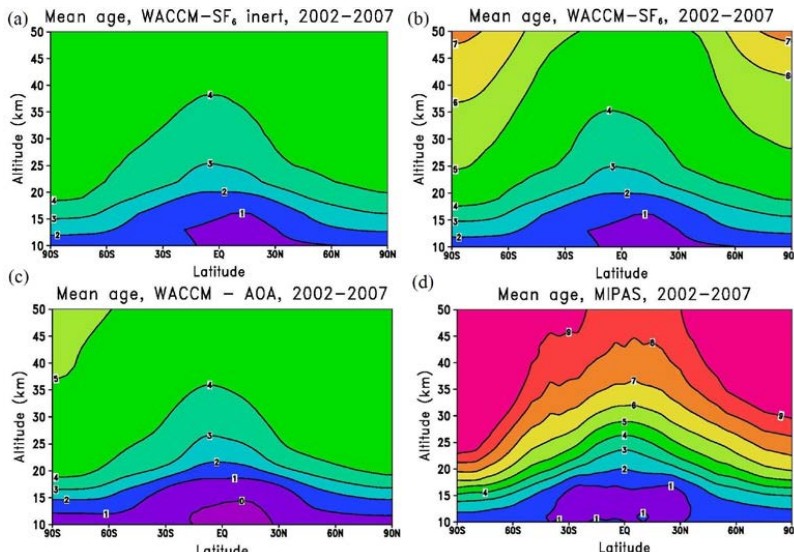

634

**Figure 7.** Annual mean age of stratospheric air (years) for the period of 2002–2007 determined

from a WACCM simulation using: (a) the inert $SF_6$ tracer; (b) the total reactive $SF_6$ tracer; (c)

the idealized AOA1 tracer. Panel (d) shows the age values derived for the same period from

our analysis of MIPAS $SF_6$ observations.





639

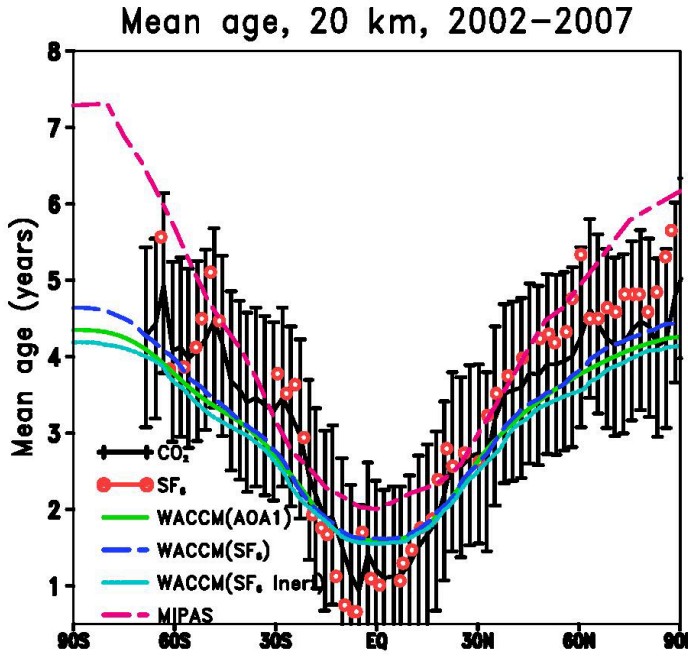

640

**Figure 8.** Mean age values at 20 km altitude derived from MIPAS satellite (dashed magenta line) and ER-2 aircraft observations ($SF_6$ red open circles, $CO_2$ black crosses) (Hall *et al.*, 1999). The error bars apply to the age derived from the ER-2 observations. Also shown is the mean age derived from WACCM tracers: reactive $SF_6$ (dashed blue line), passive $SF_6$ (light blue line) and AOA tracer (solid green line).







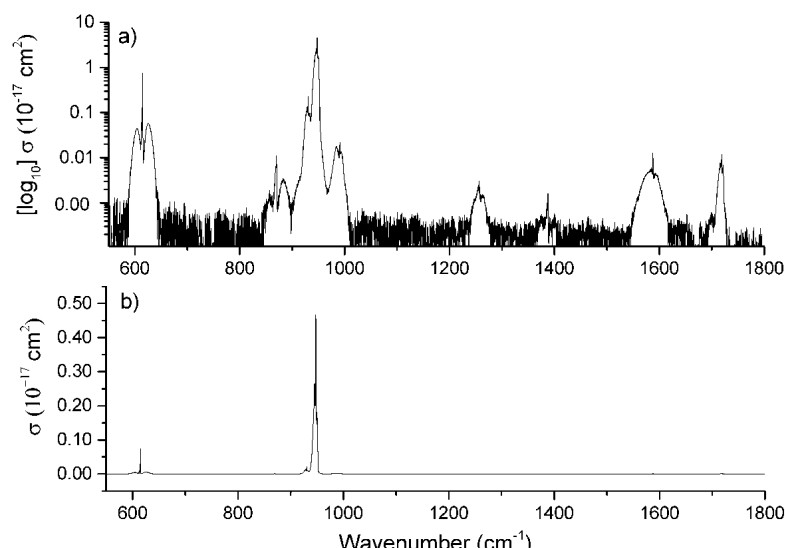



**Figure 9.** Infrared absorption spectrum of SF$_6$ at ~295 K on (a) a logarithmic y axis and (b) a
linear y axis. The logarithmic scale in panel (a) highlights the relative positions of the minor
bands.

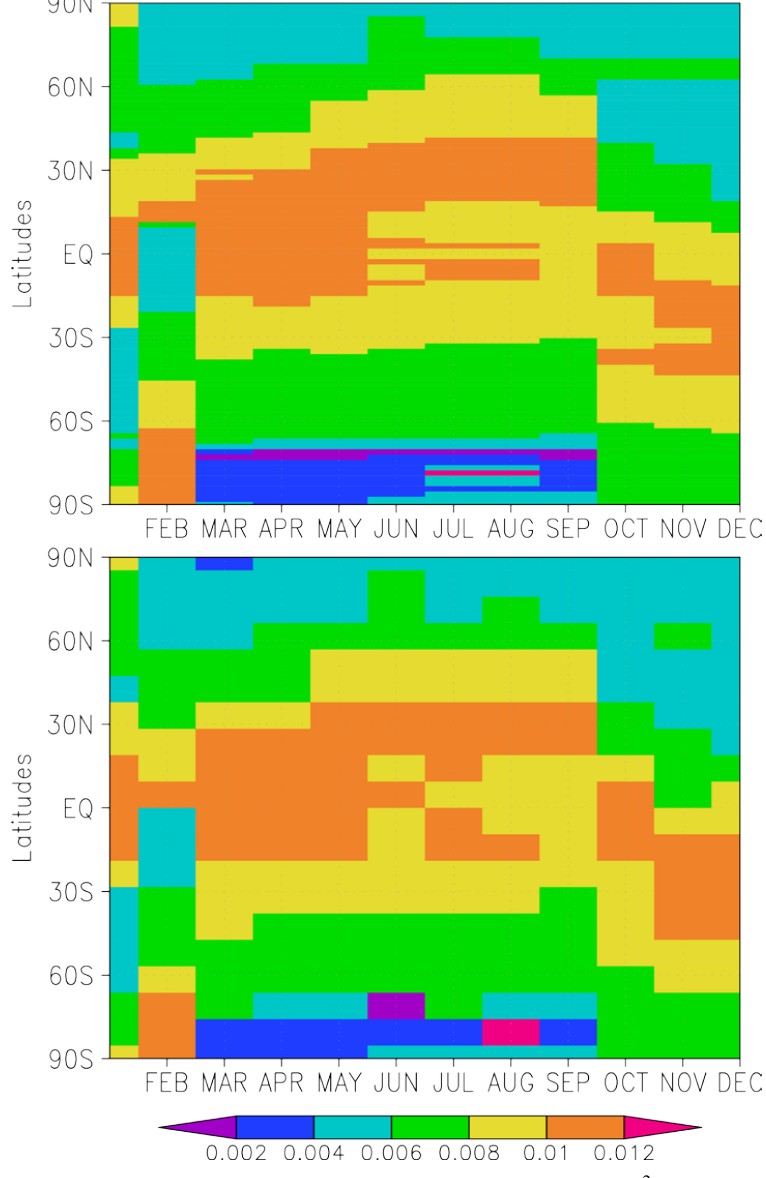


**Figure 10.** Latitude-time plots for instantaneous radiative forcing (Wm$^{-2}$) by SF$_6$ as a function
of latitude and month at (a) high latitude resolution (1.5° spacing) and (b) low latitude
resolution (9° spacing).