# Peer review of "Determination of the atmospheric lifetime and global warming potential of sulphur hexafluoride using a three-dimensional model"

_Atmospheric Chemistry and Physics, 2016_

## Referee Comment (RC1) · Anonymous Referee #1 · 22 Sep 2016

**Determination of the atmospheric lifetime and global warming potential of sulphur hexafluoride using a three dimensional model**

**The following comments rely mainly on the presentation of the results of a work of good quality. The redaction has to be revised to bring more numerical precision with the aim of the justification of the conclusions.**

**GENERAL COMMENTS**

-**All along this paper, there is a need of more precision: References, numerical values, in particular when conclusions are given. Very often, only overall appreciation is given, without documented justification.**
-**The authors make the hypothesis that the reader has "under hands" their previous papers including necessary information. In any case, it should be necessary to provide some very short reminders for not totally familiar reader with the subject and the publications of the authors.**
-**The figures are of poor qualities; it is especially difficult to read included numerical values.**

**I Introduction**

**General comments:**
- **This introduction is too long and has to be restructured.**
- **Organization of the paper with key study points should be clearly and shortly given.**
- **The sentence of lines 106-107:** *"The definitions of these radiative terms are discussed in detail in our recent publication Totterdill et al. (2016)"* **should be associated with an Appendix where definitions of ALL the radiative terms, as used all along the paper, should be recalled.**
- **Sub-paragraphs included in section: 2. (Methodology), and section 3. (Results) are misplaced or too much developed at this stage (Introduction) of the paper, i.e.:**
  *Lines 60 to 75* **should be associated with lines 311 to 334**.
  *Lines 76 to 101* **should be associated with lines 261 to 286.**
  *Lines 151 to 161* **should be associated with lines 179 to 185.**

**Other comments**
*Line 46 Surface measurements show that SF6 increased by about 7%/year during the 1980s and 1990s*
*Line 47 (Geller et al., 1997; Maiss and Brenninkmeijer, 1998).*
**No information available during the 1990's-2016's ??**

*Lines 64 and 65:* **Precise units for the quantities in (E1)**
*Lines 151 and 152:* **Quote Kovacs et al. (2016) and give a short reminder on D region and its altitude, in particular.**

**Section 2 Methodology**

**2.1    WACCM 3D model**

**General comments**

**- Paragraphs of this section to be associated with those of the introduction are identified above (may be non-exhaustive list)**
**- Some complementary brief reminder on the "Lyman-α photolysis", as used in WACCM, should be given.**

**Other comments**

*Lines 162 to 178:* **The discussion is not enough numerically documented. It relies mainly on short (non-numerical/appreciation) comments from poor quality numerical values implemented on the figures. More information should be given on the plotted values, with possibly short information tables. Please, be so kind as to revise.**

*Lines*:
185 ……………………………………………………………………..*WACCM was run for the*
*186 period 1990-2007, and the first five years were treated as spin-up. For the analysis the monthly*
*187 mean model outputs were saved and later globally averaged for the lifetime calculations*
**Those lines would be better placed after line 120.**
**Why this restricted chosen time period?**

**2.2    Infrared absorption spectrum 1and radiative forcing**

*Lines:*
*190 Previous quantitative infrared absorption spectra of SF6 have been compared in Hodnebrog et*
*191 al. (2013) (their Table 12). There are differences of ~10% between existing integrated cross*
*192 section estimates, and the measurements cover different spectral ranges.*
**Again, please, add complementary precision**

**Section 3 results**

**3.1 Global distributions of SF6 from WACCM simulations**

**General comment**

**From lines 214 to 260 revision should occur to include more numerical precision in the given conclusions, which very often only stem from figure displays (Fig. 3 and Fig. 5, in particular), with sparse or missing included numerical values.**

**Other comment**

*Line 234:  Figure 5 shows the zonal mean annual mean **SF6 distribution: which unit?***

**3.2 Atmospheric lifetime**

**General comments (already made).**

**In any case, the definition of "Atmospheric life time", should be given earlier in the text, as already suggested.**

**Complementary definition to provide, as well: "*partial* lifetime" and "*overall* lifetime".**

Other comment

*Lines:*

*300                    Finally, if we do not include the electron scaling factor to reduce the electron*
*301 density below 80 km due to negative ion formation, then the SF6 lifetime decreases to 776 years*
*302 (not shown), which is similar to the value obtained by Morris et al. (1995).*

**Please, document this sentence: "then the SF6 lifetime decreases to 776 years (not shown), which is similar to the value obtained by Morris *et al*. (1995). »**

**Section 3.4 Radiative Efficiency and Forcing**

**The references for SF6 cross-sections, as provided**

*Lines 370-372*, i.e:

*370  To determine the radiative efficiency and global warming potential of SF6, integrated cross-*
*371  sections were taken from the GEISA: 2011 Spectroscopic Database (Varanasi, 2011), the*
*372 HITRAN 2012 Molecular Spectroscopic Database (Rothman et al., 2012),.....*

**should be quoted differently. The following necessary modification in the text should be made, i.e.:**

**To determine the radiative efficiency and global warming potential of SF6, integrated cross-sections were taken from two public Molecular Spectroscopic Databases, i.e.:**
**- GEISA-2009/2011 (Jacquinet-Husson et al., 2011): data of Varanasi (2001) and Hurley (2003);**
**- HITRAN 2012 (Rothman et al., 2012): data of PNNL (Pacific Northwest National Lab) IR Database, Sharpe et al. (2004)**

- **The references should be updated accordingly, i.e.:**

➢ Jacquinet-Husson, N., Crepeau, L., Armante, R., Boutammine, C., Chédin, A., Scott, N.A., Crevoisier, C., Capelle, V., Boone, C., Poulet-Crovisier, N., Barbe, A., Campargue, A., Chris Benner, D., Benilan, Y., Bézard, B., Boudon, V., Brown, L.R., Coudert, L.H., Coustenis, A., Dana, V., Devi, V.M., Fally, S., Fayt, A., Flaud, J.-M, Goldman, A., Herman, M., Harris, G.J., Jacquemart, D., Jolly, A., Kleiner, I., Kleinböhl, A., Kwabia-Tchana, F., Lavrentieva, N., Lacome, N., Li-Hong, Xu, Lyulin, O.M., Mandin, J.-Y, Maki, A., Mikhailenko, S., Miller, C.E., Mishina, T., Moazzen-Ahmadi, N., H.S.P. Müller, A. Nikitin, J. Orphal, V. Perevalov, A. Perrin, D.T. Petkie, A. Predoi-Cross, Rinsland, C.P., Remedios,J.J., Rotger, M., Smith, M.A.H., Sung, K., Tashkun, S., Tennyson, J., Toth, R.A., Vandaele, A.-C., J. Vander Auwera, J.: The 2009 edition of the GEISA spectroscopic database. J. Quant. Spectrosc. Radiat. Transfer 112, 2395–2445, 2011; http://cds-espri.ipsl.fr/etherTypo/?id=950

➢ Varanasi P., 2001: Private communication

➢ Hurley, M.D., 2003 : Private communication.

➢ Sharpe, S.W., Johnson, T.J., Sams, R.L., Chu, P.M., Rhoderick, J.C.: Gas-Phase Databases for Quantitative Infrared Spectroscopy, Appl Spectrosc. 58(12), 1452-1461, 2004.

- **Table 4 should be revised accordingly, as well**

...................................................................................................................................................
...................................................................................................................................................

*Line 384*: Varanasi (2001): please correct Ref. in the text; no other comparison data available ?

*Lines 412-414:* References should be corrected as in lines 370-372 (see above).

*Lines 423-424*: Give reference and numerical values for: *"somewhat higher than previously published values »*

*Line 434:* Same comment as above for: *"shows a significant dependence on the solar cycle"*

---

## Referee Comment (RC2) · Anonymous Referee #2 · 21 Oct 2016

This paper is a model assessment of the atmospheric lifetime and global warming potential of sulphur hexafluoride (SF6). It relies on the use of a rather unique state-of-the-art 3-D chemistry-climate that is able to represent the key processes controlling the SF6 atmospheric cycle up to 140 km. The present work is a very substantial improvement on previous SF6 assessments. The main outputs are new updated values for SF6 atmospheric lifetime and GWP that are much more reliable than previous estimates. Using sensitivity simulations, the authors are also able to explain most the differences between previous values. Finally, they demonstrate how the SF6 loss by electron attachment in the mesosphere affect the SF6-derived mean stratospheric age-of-air (AoA) in polar regions. This should be taken into account when deriving AoA from

SF6 measurements. It is clear and well written. The methodology is solid. The title and abstract adequately represent the content of the paper. I do not have any very significant comments/corrections and therefore I recommend publication. I just provide below some minor suggestions that the authors may wish to consider.

L66: "denote the reference latitude and altitude which are chosen to be the upper tropical troposphere (latitude = 1oN, altitude = 13.9 km)". could the authors justify the choice of the latitude and altitude of the reference location, even if it is irrelevant?

L146: "Note that the SF6- anion is not modelled directly. Instead the SF6 attachment loss rate is calculated by multiplying kat by the probability of permanent destruction of the resulting SF6- (reactions of SF6- with H and HCl) to the sum of these reactions and processes which recycle SF6- to SF6 (reactions with O and O3, and photodetachment) (Morris et al., 1995)." Not very clear. And the reference give (Morris et al., 1995) cannot provide much explanations as Morris et al. assumed that the associative attachment forming SF6- does not regenerate SF6 (see L278). This could be developed in an annexe or rephrased. My understanding is that: Because of its very short lifetime, SF6- can be calculated from is photochemical steady-state expression and not transported. Therefore, replacing SF6- by its steady expression in the SF6 continuity equation, it can be shown that the net effect of electron attachment on SF6 (net because some of the SF6 lost by electron attachment is recycled back via SF6- reactions with O and O3, and photodetachment) can be calculated as the SF6 attachment loss rate multiplied by the probability of permanent destruction of the resulting SF6-; this probability is taken as the ratio of SF6- reactions rates with H and HCl over the total SF6- loss rates (i.e. SF6- reactions with H, HCl, O and O3, and photodetachment).

L251: "This model tracer can be compared to the MIPAS observations in Figure 5f, which shows that WACCM agrees reasonably well with the measurements in the lower stratosphere". "reasonably well" seems to be a bit strong and not fully consistent with the following sentence. Perhaps, something like WACCM appears to reproduce the general features of the MIPAS distribution. However, it also clear that. . .

L269: "being anti-correlated with the solar radio flux at 10.7 cm". Even though most solar activity indices are strongly correlated, the more meaningful indices for electron density are solar fluxes at much shorter wavelengths (e.g. X-rays, Lyman alpha). Make the statement more general: being anti-correlated with solar activity, for example as measured by the radio flux at 10.7 cm. . .

L228: "As is clear from Figure 3, the model simulation and satellite observations agree within the atmospheric variability, which becomes relatively large above 30 km especially at high latitudes, although the model is systematically larger than the observations above 20 km.". The two parts of this statement seem to be partly contradictory or, at least, confusing. Usually, when observations and models are compared, it can be done either on the mean (climatology) or the variability (standard deviation around the mean). For example, a model can reproduce the observed variability but have a bias on the mean. And the reverse is also possible. I guess the statement "the model simulation and satellite observations agree within the atmospheric variability" is meant to be applied to the mean. In that case, as stated in the second part of the statement, there is a clear systematic bias in the mean here and the fact that there is a large variability (standard deviation) with bars covering model-calculated is not relevant here. If the targeted quantity for the comparison is the mean, the relevant uncertainty is the error on the mean (which depends on the sigma but not only, it depends also on the number of profiles) and the same thing should be done on the model-calculated profiles. If observed and model profiles with errors on the means are partly covering each other, one can then claim that obs and model agree within the errors. But I guess that is not the case here. I would suggest to state simply that the variability is high and that there is a systematic bias. If the authors want to get into the variability, they can calculate in their model. I don't think it would bring anything valuable here.

L287: "Our estimated partial lifetime of SF6 due to photolysis for the SF6 tracer which includes all loss processes is 48,000 yr". SF6 due to photolysis but SF6 tracer includes all loss processes? Not clear. Rephrase.

L411: "They established the range of published radiative efficiencies for SF6 to be 0.59 – 0.68 Wm-2ppbv-1, with a mean value of 0.56 Wm-2 ppbv-1 " The mean value does not correspond to the range. For instance, just above is cited the value of Myhre et al., (2013): 0.57 Wm-2 ppbv-1. The range needs to be corrected.

The quality of Figure 8 can be improved.

---

## Author Comment (AC1) · 5 Dec 2016

**Determination of the atmospheric lifetime and global warming potential of sulphur hexafluoride using a three dimensional model**

Kovacs et al.

We thank the reviewers for their comments. These comments are reproduced below in *italics*, followed by our responses in **bold red**.

**Reviewer 1**

*The following comments rely mainly on the presentation of the results of a work of good quality. The redaction has to be revised to bring more numerical precision with the aim of the justification of the conclusions.*

**General Comments**

*All along this paper, there is a need of more precision: References, numerical values, in particular when conclusions are given. Very often, only overall appreciation is given, without documented justification.*

*The authors make the hypothesis that the reader has "under hands" their previous papers including necessary information. In any case, it should be necessary to provide some very short reminders for not totally familiar reader with the subject and the publications of the authors.*

*The figures are of poor qualities; it is especially difficult to read included numerical values.*

**We have replotted the figures.**

**I Introduction**

**General comments:**

*This introduction is too long and has to be restructured.*
*Organization of the paper with key study points should be clearly and shortly given.*

**See responses to specific comments below.**

The sentence of lines 106-107: *"The definitions of these radiative terms are discussed in detail in our recent publication Totterdill et al. (2016)"* should be associated with an Appendix where definitions of ALL the radiative terms, as used all along the paper, should be recalled.

**OK. We have added Appendix A which defines the relevant radiative and climate metrics used.**

Sub-paragraphs included in section: 2. (Methodology), and section 3. (Results) are misplaced or too much developed at this stage (Introduction) of the paper, i.e.:
*Lines 60 to 75* should be associated with lines 311 to 334.
*Lines 76 to 101* should be associated with lines 261 to 286.
*Lines 151 to 161* should be associated with lines 179 to 185.

**We thank the reviewer for this suggestion but we feel that the present arrangement of material is logical and appropriate. We note that Reviewer 2 found the paper 'clear and well written' as it is.**

**Other comments**

*Line 46 Surface measurements show that SF6 increased by about 7%/year during the 1980s and 1990s  Line 47 (Geller et al., 1997; Maiss and Brenninkmeijer, 1998).*
*No information available during the 1990's-2016's ??*

**Yes, this statement was outdated. We have added a reference to Dlugokencky et al. (2016) and updated the %/year increase rate.**

*Lines 64 and 65: Precise units for the quantities in (E1).*

**The equation is general. AoA will have the same units as t, and can be any measure of time. We do not need to give specific units.**

*Lines 151 and 152: Quote Kovacs et al. (2016) and give a short reminder on D region and its altitude, in  particular.*

**OK. We have added the following sentence: "The *D* region is the lowest part of the ionosphere, extending from about 60 to 85 km. It is characterized by the appearance of cluster ions (e.g. proton hydrates $H^+.(H_2O)_n$, where $n \leq 6$) and negative ions (e.g. $O_2^-$, $CO_3^-$ and $NO_3^-$) rather than free electrons. These species predominate because the atmospheric pressure is high enough to facilitate the three-body attachment of ligand species like $H_2O$ to positive ions, and electrons to neutral molecules."**

**Section 2 Methodology**

**2.1 WACCM 3D Model**

**General Comments**

*Paragraphs of this section to be associated with those of the introduction are identified above (may be   non-exhaustive list).*

**See response above. We have left the order of material unchanged.**

*Some complementary brief reminder on the "Lyman-α photolysis", as used in WACCM, should be  given.*

**We already give information on the Lyman-a scheme in WACCM by giving the fluxes used and the value for the SF6 cross section.**

**Other comments**

*Lines 162 to 178: The discussion is not enough numerically documented. It relies mainly on short (non- numerical/appreciation) comments from poor quality numerical values implemented on the figures. More information should be given on the plotted values, with possibly short information tables. Please, be so kind as to revise.*

**We have added in some references to numbers in the plots.**

*Lines 185 to 187: "WACCM was run for the period 1990-2007, and the first five years were treated as spin-up. For the analysis the monthly mean model outputs were saved and later globally averaged for the lifetime calculations"*
*Those lines would be better placed after line 120.*
*Why this restricted chosen time period?*

**The first paragraph of Section 2.1 gives general information about the model. Specific information about the experiments performed (number of tracers, duration) is given in this paragraph (old line 179 onwards). Therefore, we feel that this sentence is in the correct place. WACCM is quite expensive to run and so we chose a time period which was sufficient to investigate the SF6 lifetime over a whole solar cycle. The period from mid 1990s onwards was optimum for comparison with observations (e.g. MIPAS).**

**2.2 Infrared absorption spectrum 1and radiative forcing**

*Lines 190-192: "Previous quantitative infrared absorption spectra of SF6 have been compared in Hodnebrog et al. (2013) (their Table 12). There are differences of ~10% between existing integrated cross section estimates, and the measurements cover different spectral ranges."*
*Again, please, add complementary precision*

**The review paper Hodnebrog et al. (2013) does not discuss the precision of the IR spectra measurements, nor is it comprehensively covered in the underlying papers. Therefore, it is difficult to assess. Precision also varies greatly with wavelength and methodology and generally assumed to be better than systematic bias between measurements. It is the spread of previous integrated cross section measurements that we deem important to consider here, so feel adding to the text would only confuse the reader and we choose to leave as is.**

**Section 3 Results**

**3.1 Global distributions of SF6 from WACCM simulations**

**General comment**

*From lines 214 to 260 revision should occur to include more numerical precision in the given conclusions, which very often only stem from figure displays (Fig. 3 and Fig. 5, in particular), with sparse or missing included numerical values.*

**We have added in numerical values throughout this section.**

**Other comment**

*Line 234: Figure 5 shows the zonal mean annual mean SF6 distribution: which unit?*

**The units of pptv are stated in the figure caption.**

**3.2 Atmospheric lifetime**

**General comments (already made).**

*In any case, the definition of "Atmospheric lifetime", should be given earlier in the text, as already suggested.*
*Complementary definition to provide, as well: "partial lifetime" and "overall lifetime".*

**Atmospheric lifetime has been defined when first used in Section 1. In Section 3.2 we explain the terms partial and overall lifetimes when first used.**

**Other comment**
*Lines 300-302: "Finally, if we do not include the electron scaling factor to reduce the electron density below 80 km due to negative ion formation, then the SF6 lifetime decreases to 776 years (not shown), which is similar to the value obtained by Morris et al. (1995)".*
*Please, document this sentence: "then the SF6 lifetime decreases to 776 years (not shown), which is similar to the value obtained by Morris et al. (1995)."*

**We have added the value obtained by Morris et al. to the text.**

**Section 3.4 Radiative Efficiency and Forcing**

*The references for SF6 cross-sections, as provided in lines 370-372, "To determine the radiative efficiency and global warming potential of SF6, integrated cross-sections were taken from the GEISA: 2011 Spectroscopic Database (Varanasi, 2011), the HITRAN 2012 Molecular Spectroscopic Database (Rothman et al., 2012)" should be quoted differently. The following necessary modification in the text should be made, i.e.:*

*To determine the radiative efficiency and global warming potential of SF6, integrated cross-sections were taken from two public Molecular Spectroscopic Databases, i.e.:*
- *GEISA-2009/2011 (Jacquinet-Husson et al., 2011): data of Varanasi (2001) and Hurley (2003);*
- *HITRAN 2012 (Rothman et al., 2012): data of PNNL (Pacific Northwest National Lab) IR Database, Sharpe et al. (2004)*

*The references should be updated accordingly, i.e.:*

- *Jacquinet-Husson, N., Crepeau, L., Armante, R., Boutammine, C., Chédin, A., Scott, N.A., Crevoisier, C., Capelle, V., Boone, C., Poulet-Crovisier, N., Barbe, A., Campargue, A., Chris Benner, D., Benilan, Y., Bézard, B., Boudon, V., Brown, L.R., Coudert, L.H., Coustenis, A., Dana, V., Devi, V.M., Fally, S., Fayt, A., Flaud, J.-M, Goldman, A., Herman, M., Harris, G.J., Jacquemart, D., Jolly, A., Kleiner, I., Kleinböhl, A., Kwabia-Tchana, F., Lavrentieva, N., Lacome, N., Li-Hong, Xu, Lyulin,*

*O.M., Mandin, J.-Y, Maki, A., Mikhailenko, S., Miller, C.E., Mishina, T., Moazzen-Ahmadi, N., H.S.P. Müller, A. Nikitin, J. Orphal, V. Perevalov,Perrin, D.T. Petkie, A. Predoi-Cross, Rinsland, C.P., Remedios,J.J., Rotger, M., Smith, M.A.H., Sung, K., Tashkun, S., Tennyson, J., Toth, R.A., Vandaele, A.-C., J. Vander Auwera, J.: The 2009 edition of the GEISA spectroscopic database. J. Quant. Spectrosc. Radiat. Transfer 112, 2395–2445, 2011;* http://cds-espri.ipsl.fr/etherTypo/?id=950

- *Varanasi P., 2001: Private communication*
- *Hurley, M.D., 2003: Private communication.*
- *Sharpe, S.W., Johnson, T.J., Sams, R.L., Chu, P.M., Rhoderick, J.C.: Gas-Phase Databases for Quantitative Infrared Spectroscopy, Appl Spectrosc. 58(12), 1452-1461, 2004.*

**OK. The references and text have been updated. The personal communications will need to conform to the journal regulations.**

*Table 4 should be revised accordingly, as well*

**OK.**

*Line 384: Varanasi (2001): please correct Ref. in the text; no other comparison data available?*

**OK, changed to 2001.**

*Lines 412-414: References should be corrected as in lines 370-372 (see above).*

**OK, done.**

*Lines 423-424: Give reference and numerical values for: "somewhat higher than previously published values"*

**This text follows on from the previous sentences which compare the values in Table 6. We have added 'the' to make it clearer.**

*Line 434:* Same comment as above for: *"shows a significant dependence on the solar cycle"*

**OK. At this line we have inserted the magnitude of this dependency**

---

## Author Comment (AC2) · 5 Dec 2016

**Determination of the atmospheric lifetime and global warming potential of sulphur hexafluoride using a three dimensional model**

Kovacs et al.

We thank the reviewers for their comments. These comments are reproduced below in *italics*, followed by our responses in **bold red**.

**Reviewer 2**

*This paper is a model assessment of the atmospheric lifetime and global warming potential of sulphur hexafluoride (SF6). It relies on the use of a rather unique state-of-the-art 3-D chemistry-climate that is able to represent the key processes controlling the SF6 atmospheric cycle up to 140 km. The present work is a very substantial improvement on previous SF6 assessments. The main outputs are new updated values for SF6 atmospheric lifetime and GWP that are much more reliable than previous estimates. Using sensitivity simulations, the authors are also able to explain most the differences between previous values. Finally, they demonstrate how the SF6 loss by electron attachment in the mesosphere affect the SF6-derived mean stratospheric age-of-air (AoA) in polar regions. This should be taken into account when deriving AoA from SF6 measurements. It is clear and well written. The methodology is solid. The title and abstract adequately represent the content of the paper. I do not have any very significant comments/corrections and therefore I recommend publication. I just provide below some minor suggestions that the authors may wish to consider.*

**We thank the reviewer for his/her comments**.

*L66: "denote the reference latitude and altitude which are chosen to be the upper tropical troposphere (latitude = 1oN, altitude = 13.9 km)". Could the authors justify the choice of the latitude and altitude of the reference location, even if it is irrelevant?*

**The general choice is the upper tropical troposphere as stated. Within the model we need to choose a specific gridpoint location, and so we chose the point indicated. We have added some words to say that this is a model gridpoint within the required region.**

*L146: "Note that the SF6- anion is not modelled directly. Instead the SF6 attachment loss rate is calculated by multiplying $k_{at}$ by the probability of permanent destruction of the resulting SF6- (reactions of SF6- with H and HCl) to the sum of these reactions and processes which recycle SF6- to SF6 (reactions with O and O3, and photodetachment) (Morris et al., 1995)." Not very clear. And the reference give (Morris et al., 1995) cannot provide much explanations as Morris et al. assumed that the associative attachment forming SF6- does not regenerate SF6 (see L278). This could be developed in an annexe or rephrased.*
*My understanding is that: Because of its very short lifetime, SF6- can be calculated from is photochemical steady-state expression and not transported. Therefore, replacing SF6- by its steady expression in the SF6 continuity equation, it can be shown that the net effect of electron attachment on SF6 (net because some of the SF6 lost by electron attachment is recycled back via SF6- reactions with O and O3, and photodetachment) can be calculated as the SF6 attachment loss rate multiplied by the probability of permanent destruction of the resulting SF6-; this probability is taken as the ratio of SF6- reactions rates with H and HCl over the total SF6- loss rates (i.e. SF6- reactions with H, HCl, O and O3, and photodetachment).*

**OK. We have rewritten this section and deleted the reference to Morris et al. here.**

*L251: "This model tracer can be compared to the MIPAS observations in Figure 5f, which shows that WACCM agrees reasonably well with the measurements in the lower stratosphere". "reasonably well" seems to be a bit strong and not fully consistent with the following sentence. Perhaps, something like WACCM appears to reproduce the general features of the MIPAS distribution. However, it also clear that...*
**OK. We have rephrased the text as suggested.**

*L269: "being anti-correlated with the solar radio flux at 10.7 cm". Even though most solar activity indices are strongly correlated, the more meaningful indices for electron density are solar fluxes at much shorter wavelengths (e.g. X-rays, Lyman alpha). Make the statement more general: being anti-correlated with solar activity, for example as measured by the radio flux at 10.7 cm...*

**OK. The suggested text has been added to make it more general**.

*L228: "As is clear from Figure 3, the model simulation and satellite observations agree within the atmospheric variability, which becomes relatively large above 30 km especially at high latitudes, although the model is systematically larger than the observations above 20 km". The two parts of this statement seem to be partly contradictory or, at least, confusing. Usually, when observations and models are compared, it can be done either on the mean (climatology) or the variability (standard deviation around the mean). For example, a model can reproduce the observed variability but have a bias on the mean. And the reverse is also possible. I guess the statement "the model simulation and satellite observations agree within the atmospheric variability" is meant to be applied to the mean. In that case, as stated in the second part of the statement, there is a clear systematic bias in the mean here and the fact that there is a large variability (standard deviation) with bars covering model-calculated is not relevant here. If the targeted quantity for the comparison is the mean, the relevant uncertainty is the error on the mean (which depends on the sigma but not only, it depends also on the number of profiles) and the same thing should be done on the model-calculated profiles. If observed and model profiles with errors on the means are partly covering each other, one can then claim that obs and model agree within the errors. But I guess that is not the case here. I would suggest to state simply that the variability is high and that there is a systematic bias. If the authors want to get into the variability, they can calculate in their model. I don't think it would bring anything valuable here.*

**OK. We have rephrased this to compare the means and then mention that the variability is large.**

*L287: "Our estimated partial lifetime of SF6 due to photolysis for the SF6 tracer which includes all loss processes is 48,000 yr". SF6 due to photolysis but SF6 tracer includes all loss processes? Not clear. Rephrase.*

**The text meant that if we consider the SF6 tracer which includes all loss processes, and diagnose the contributions from the different terms, we derive a partial lifetime wrt photolysis of 48,000 yrs. This has been rewritten.**

*L411: "They established the range of published radiative efficiencies for SF6 to be 0.59 – 0.68 Wm-2ppbv-1, with a mean value of 0.56 Wm-2 ppbv-1" The mean value does not correspond to the range. For instance, just above is cited the value of Myhre et al., (2013): 0.57 Wm-2 ppbv-1. The range needs to be corrected.*

**OK, apologies. The 0.59 was a typo and has been corrected to 0.49.**

*The quality of Figure 8 can be improved.*

**Figure 8 has been replotted and is now much clearer.**